# Report of Adverse Effects Following Population-Wide COVID-19 Vaccination: A Comparative Study between Six Different Vaccines in Baja-California, Mexico

**DOI:** 10.3390/vaccines10081196

**Published:** 2022-07-27

**Authors:** Cesar A. Mendez-Lizarraga, Enrique Chacon-Cruz, Ricardo Carrillo-Meza, Néstor Saúl Hernández-Milán, Leslie C. Inustroza-Sánchez, Diego F. Ovalle-Marroquín, Jesús René Machado-Contreras, Omar Ceballos Zuñiga, Verónica Bejarano-Ramírez, Cipriano Aguilar-Aguayo, Adrián Medina-Amarillas, Santa Elizabeth Ceballos-Liceaga, Oscar E. Zazueta

**Affiliations:** 1Departamento de Epidemiología, Secretaría de Salud de Baja California, Mexicali 21000, Mexico; cinustroza@uabc.edu.mx; 2Departamento de Infectología Pediátrica, Hospital General de Tijuana, Tijuana 22000, Mexico; echacon88@hotmail.com; 3Unidad Ciencias de la Salud, Universidad Autónoma de Baja California, Mexicali 21376, Mexico; carrillo.ricardo@uabc.edu.mx; 4Dirección General de Servicios de Salud del Estado de Baja California, Secretaría de Salud de Baja California, Mexicali 21000, Mexico; nshernandez@salud.gob.mx (N.S.H.-M.); cipriano.aguilar@saludbc.gob.mx (C.A.-A.); jmedinaa@baja.gob.mx (A.M.-A.); 5Dirección de Enseñanza e Investigación, Secretaría de Salud de Baja California, Mexicali 21000, Mexico; dfovalle@issstecali.gob.mx; 6Facultad de Medicina, Universidad Autónoma de Baja California, Mexicali 21000, Mexico; rene.machado@uabc.edu.mx; 7Departamento de Medicina Interna, Hospital General de Mexicali, Mexicali 21000, Mexico; neumologo.oc@breathbaja.com; 8Laboratorio Estatal de Salud Pública, Secretaría de Salud de Baja California, Mexicali 21010, Mexico; vbejarano@salud.gob.mx; 9Dirección General de Epidemiología, Secretaría de Salud, Gobierno de México, Ciudad de México 01480, Mexico; elizabeth.ceballos@salud.gob.mx

**Keywords:** adverse events, vaccines, COVID-19, Mexico, epidemiological surveillance

## Abstract

After emergency authorization, different COVID-19 vaccines were administered across Mexico in 2021, including mRNA, viral vector, and inactivated platform vaccines. In the state of Baja-California, 3,516,394 doses were administered, and 2285 adverse events (AE) were registered in the epidemiological surveillance system in 2021. Incidence rates per 100,000 doses were calculated for total, mild (local and systemic), and severe AE for each vaccine. Symptoms were compared between mRNA and viral vector/inactivated virus vaccines. The overall incidence rate for all AE was 64.98 per 100,000 administered doses; 79.05 AE per 100,000 doses for mRNA vaccines; and 56.9 AE per 100,000 doses for viral vector/inactivated virus vaccine platforms. AE were at least five times higher in recipients of the AstraZeneca vaccine from the Serum Institute of India (AZ from SII). Local injection site symptoms were more common in mRNA vaccines while systemic were more prevalent in viral vector/inactivated virus vaccines. Severe AE rates were similar across all administered vaccines (0.72–1.61 AE per 100,000 doses), except for AZ from SII, which documented 12.6 AE per 100,000 doses. Among 32 hospitalized severe cases, 28 (87.5%) were discharged. Guillain–Barré Syndrome was the most common serious AE reported (*n* = 7). Adverse events rates differed among vaccine manufacturers but were consistent with clinical trials and population-based reports in the literature.

## 1. Introduction

On 31 December 2019, the Municipal Health Commission of Wuhan (Hubei Province, China) reported a cluster of pneumonia cases in the city; subsequently, it was determined that they were caused by a novel coronavirus, identified and named as “severe acute respiratory syndrome coronavirus” (SARS-CoV-2) or COVID-19, an acronym for “coronavirus disease of 2019” [1]. After a month, COVID-19 spread from the People’s Republic of China to 20 other countries. On 30 January 2020, following the Emergency Committee’s recommendations, the WHO Director-General declared that the outbreak constituted a Public Health Emergency of International Concern [2], which was later characterized as a pandemic on 11 March 2020 [3].

The race to make a COVID-19 vaccine started, and lessons learned from pre-clinical and clinical data of previous vaccine trials (SARS-CoV and MERS-CoV) [4], along with the early publication of the viral sequence of SARS-CoV-2, enabled work on a vaccine within weeks of China’s initial notification to the WHO on 31 December 2019 [5]. Different vaccine platforms were explored, which were divided into three groups: 1. “traditional” platforms, such as inactivated virus vaccines or live-attenuated virus vaccines; 2. those that were recently licensed, such as recombinant protein vaccines and viral vector-based vaccines; and 3. platforms with no previous authorized vaccines for use, such as mRNA- and DNA/RNA-based vaccines [6].

The process from viral genome sequencing to the development of a vaccine took less than a year to develop, when the Ad5-nCoV vaccine was approved on 25 June 2020 by China’s Central Military Commission. Later, the Gam-COVID-Vac Lyo/Sputnik vaccine was approved on 24 August 2020 by Russia [7]. As of 22 April 2022, forty-eight COVID-19 vaccines were under development in Phase 1 trials, sixty-seven in Phase 2 trials, and sixty-one in Phase 3 trials, while thirty-eight had received emergency approval around the globe [8].

Baja-California is a northwestern state in Mexico that shares the border with the state of California, United States of America (USA). As of 2020, it had 3.7 million habitants (50.4% men, with a mean age of 30 years), and it comprises two of Mexico’s largest metropolitan areas: Tijuana and Mexicali (1.9 and 1.04 million habitants, respectively) [9]. Both cities total seven Mexico–USA land entry ports from which more than 21 million pedestrians, 31 million personal vehicles, and 1.4 million merchandise trucks had crossed the border in 2019 [10]. Additionally, the state is also an economic hub; in 2019 it had purchases worth USD 14.4 billion in merchandise from the USA and had sales of USD 32.7 billion to the same country, which was also the main foreign investor (USD 20.5 billion between 1999–2021) [9].

On 11 December 2020, the first emergency approval in Mexico was granted to the BNT162b2 mRNA COVID-19 vaccine by the Federal Commission for Protection Against Sanitary Risks (COFEPRIS, by its acronym in Spanish), initiating the national vaccination campaign on 24 December 2020 [11,12]. By 2 March 2022, COFEPRIS had granted emergency approval to a total of ten COVID-10 vaccines [13]. Based on recommendations by a technical workgroup, the Mexican vaccination campaign started with priority groups and gradually included different population age groups [11] (Figure 1), thus demanding an increased role in epidemiological surveillance for detection, investigation, and analysis of adverse events (AE) supposedly attributed to vaccination or immunization, defined as “clinical manifestations or medical events that occur after vaccinations and are supposedly attributed to vaccination or immunization” by the General Directorate of Epidemiology of Mexico [14]. As of 31 March 2022, more than 130 million doses were administered across the country, with 35,857 non-severe and 1078 severe AE notified in Mexico. In the state of Baja-California, more than 3 million doses were administered in 2021, with 2224 mild AE and 39 severe AE documented [15].

The objective of the present study is to describe and compare the AE reported during the COVID-19 vaccination campaign in the state of Baja-California, Mexico, in 2021, which included the following six different vaccines: BNT162b2 (Pfizer-BioNTech), ChAdOx1 nCoV-19/AZD-1222 (AstraZeneca, AZ from Oxford), ChAdOx1 nCoV-19/COVISHIELD (Serum Institute of India PVT, AZ from SII), CoronaVac (Sinovac), Ad5-nCoV (CansinoBIO), and Ad26.CoV2.S (Johnson & Johnson/Janssen, J&J).

## 2. Materials and Methods

### 2.1. Study Design

We analyzed data from the epidemiological surveillance system of adverse events (AE) supposedly attributed to vaccination or immunization from January through December of 2021, following COVID-19 vaccination in Baja-California, Mexico.

### 2.2. Setting and Participants

Between January and December of 2021, a total of 3,516,394 doses of COVID-19 vaccines were administered in Baja-California, Mexico (Figure 2). We describe and compare the incidence rates of AE from six different COVID-19 vaccines that were applied in the State.

Case definitions were established nationally by the Directorate General of Epidemiology (DGE), and the surveillance methodology followed national standards. A mild adverse event (MAE) was defined as an individual of any age or gender presenting local and/or systemic clinical manifestations in the first 30 days after immunization and supposedly attributed to vaccination or immunization. Additionally, the event in question did not put the life of the patient at risk, it disappeared with or without symptomatic treatment, and was not a cause of hospitalization or long-term disability [14].

A severe adverse event (SAE) was defined as an individual of any age who, in the first 30 days following vaccination or immunization, presented one or more of the following clinical manifestations:Caused the death of the patient.Placed the life of the patient at imminent risk.Caused disability or persistent and significant impairment.Required hospitalization or extended length of hospital stay.

Examples of SAE include seizures, severe dehydration, anaphylactic shock, acute flaccid paralysis, encephalitis, intracranial hemorrhage, profuse diarrhea, or persistent vomit.

All reports of AE following COVID-19 vaccination in the State were included in the study. The research protocol was approved by the Institutional Review Board of Tijuana General Hospital, Mexico (approval no. CONBIOETICA-02-CEI-001-20170526).

### 2.3. Measurement Instrument and Variables

The national case form for reporting AE after vaccination for COVID-19 in Mexico includes information on the history of allergies, pregnancy status, information on the vaccine (manufacturer, number of doses, date of application, expiration date, vaccine lot number), history of COVID-19 infection, a 53-item checklist of clinical signs and symptoms, the classification of the AE, and information on the site of vaccine application and the medical unit that reports the AE [14].

### 2.4. Statistical Analysis

AE incidence rates were calculated per 100,000 doses administered. Quantitative variables were described as means and standard deviations, while categorical variables were presented as absolute frequencies and proportions. Exploratory analyses were conducted to evaluate differences between the clinical manifestations of recipients of mRNA vaccines vs. viral vector and inactivated virus vaccines. Logistic regression was employed to explore an association between the type of vaccine platform and the development of SAE. Odds ratios adjusted for age and gender were calculated for this exploratory analysis. For all tests, *p*-values < 0.05 were considered statistically significant. All statistical analyses were conducted using RStudio (Version 1.4.1103, 2009-2021 Rstudio, PBC).

## *3.* Results

### 3.1. Adverse Events Rates by Vaccine Manufacturer

A total of 2326 AE were reported through the state epidemiological surveillance system following vaccination out of 3,516,394 COVID-19 doses administered during 2021. We excluded 41 cases who reported being vaccinated in the United States of America or were under age 12, as the latter were not part of the state vaccination policy, with a total of 2285 AE included in this study (Figure 2). Mild adverse events summed 2253 (98.59% of all AE), and the remaining 32 (1.4%) were SAE; 28 (1.22%) of which were discharged from hospitalization, and 4 (0.17%) resulted in deaths.

From 2,233,602 administered doses of viral vector and inactivated virus vaccines, a total of 1271 (0.05%) AE were reported vs 1014 (0.07%) AE documented across 1,282,792 doses of mRNA (Pfizer-BioNTech) vaccines (Table 1). The overall incidence rate of both MAE and SAE was 64.98 per 100,000 doses; with the highest incidence rate observed among AZ from SII vaccine recipients (474.59 AE per 100,000 doses), followed by CansinoBIO (84.38 AE per 100,000 doses), Pfizer-BioNTech (79.05 AE per 100,000 doses), J&J (65.66 AE per 100,000 doses), AZ from Oxford (35.71 AE per 100,000 doses), and Sinovac (15.28 AE per 100,000 doses). Compared to the AE incidence rate among mRNA vaccine recipients (79.05 AE per 100,000 doses), statistically significant differences (*p* < 0.0001) were observed among all viral vector and inactivated virus vaccines, except in CansinoBIO vaccine recipients (*p* = 0.61).

Two-sided hypothesis testing for the equality of proportions was carried out comparing the AE rates for each vaccine manufacturer per 100,000 applied doses against the mRNA AE rate (Table 1).

MAE represented 98.59% (*n* = 2253) of all AE, and the highest MAE incidence rate was observed among recipients of AZ from SII (461.99 AE per 100,000 doses), followed by CansinoBIO (83.34 AE per 100,000 doses), Pfizer-BioNTech (78.27 AE per 100,000 doses), J&J (64.94 AE per 100,000 doses), AZ from Oxford (34.34 AE per 100,000 doses), and Sinovac (13.67 AE per 100,000 doses). SAE constituted 1.4% (*n* = 32) of all reported events, with an overall incidence rate of 0.91 AE per 100,000 doses. AZ from SII recipients also experienced the highest SAE incidence rate (12.6 AE per 100,000 doses) compared to the other vaccine recipients: Sinovac (1.61 AE per 100,000 doses), CansinoBIO (1.04 AE per 100,000 doses), AZ from Oxford (0.82 AE per 100,000 doses), Pfizer-BioNTech (0.78 AE per 100,000 doses), and J&J (0.72 AE per 100,000 doses).

### 3.2. Demographic Characteristics and Reported Symptoms

The mean age among patients who received viral vector/inactivated virus was 35.42 years (SD 12.98) and 40.5 years (SD 12.29) for patients who received mRNA vaccines (Table 2). Overall, women represented the majority of reported AE: 66.38% and 64.2% in viral vector/inactivated virus and mRNA groups, respectively (*p* < 0.0001). Of those women, 1.57% were pregnant in the viral vector group vs. 0.89% in the mRNA group (*p* = 0.11).

History of allergic reactions differed between viral vector/inactivated virus and mRNA recipients (12.03% vs. 17.06%, *p* < 0.001). Specific allergic reactions, such as food (1.1% vs. 3.06%) and drugs (6.37% vs. 9.86%), were also different between both groups. The most frequently reported symptoms in viral vector/inactivated virus and mRNA recipients were headache (81.9% vs. 71.3%, respectively), myalgias/arthralgias (71.99% vs. 53.55%), fever/hyperthermia (54.84% vs. 36.89%), chills/diaphoresis (49.49% vs. 32.35%), and injection site reactions (47.99% vs. 59.37%), all of which had statistical differences in its frequency between the two groups (*p* < 0.0001). Local injection site reactions included nodule or induration, pain, erythema, edema, cellulitis, pruritus, abscess formation, and increased local temperature. Other symptoms that differed in frequency between viral vector/inactivated virus and RNA recipients were gastrointestinal symptoms (47.13% vs. 41.62%), rhinorrhea (13.3 vs. 17.06%), palpitations/tachycardia (7.87% vs. 10.75%), dyspnea/difficult breathing (7.63 vs. 5.42%), and peripheral neurologic symptoms (2.99% vs. 5.13%). Reported gastrointestinal symptoms included nausea, vomiting, abdominal pain, diarrhea, gastrointestinal bleeding, and intussusception. Peripheral neurologic symptoms included paresthesia, dysesthesia, weakness, paresis, and Bell’s palsy.

### 3.3. Severe Adverse Events

The incidence rate of SAE was 0.78 AE per 100,000 doses for mRNA vaccines (*n* = 1,282,792) and 0.09 SAE per 100,000 doses for viral vector/inactivated virus vaccines recipients (*n* = 2,233,602). Of all reported AE (*n* = 2285), severe ones comprised 1.4%. The most frequent clinical diagnosis from SAE cases was Guillain–Barré syndrome (21.8%) (Table 3), acute polyradiculopathy (9.37%), transverse myelitis (6.25%), and acute myocardial infarction (6.25%). Out of 32 SAE, twenty-eight cases (87.5%) were discharged from hospitalization and four died (12.5%). The average time of symptom onset was 8.7 h after vaccine administration, and the average length of stay was 5.1 days among admitted patients. Causality evaluation revealed that most of these events were (A1) related to the vaccine (59.37%, *n* = 19), followed by (B) undetermined (9.37%, *n* = 3) and (A1/B) related to vaccine/undetermined (9.37%, *n* = 3), and (D/E) associated to inherent characteristics of the vaccinated individual/unclassifiable (6.25%, *n* = 2). Each of the following causality evaluations, (C) inconsistent with vaccination, (D) due to inherent characteristics of the vaccinated individual, (A1/A3) related to the vaccine/related to vaccine quality defect, and (A1/C) related to the vaccine/inconsistent with vaccination, had a frequency of one (3.12%, *n* = 1). Three out of four deaths (75%) were determined to be related (A1) to the vaccine.

An exploratory analysis showed that receiving an mRNA vaccine was not associated to higher odds of developing a SAE when compared to viral vector and inactivated virus vaccine recipients (OR 0.50, 95%CI 0.23–1.07, *p* = 0.07), after adjusting for age and gender. Age was found to be an important predictor of reporting a SAE in this exploratory analysis (OR 1.03, 95%CI 1.01–1.06, *p* = 0.01). 

## 4. Discussion

Our study describes and compares mild and severe adverse events following the immunization for COVID-19 between mRNA vaccine recipients and other vaccine platforms (viral vector and inactivated virus vaccines) during the population-wide state campaign in Baja-California, Mexico (January to December 2021). We calculated AE incidence rates per 100,000 doses for each applied vaccine, and we compared symptoms between mRNA vaccines and other platforms. Additionally, we described severe AE in more detail.

The overall incidence rate for all AE was 64.98 per 100,000 administered doses: 79.05 AE per 100,000 doses for mRNA vaccines and 56.9 AE per 100,000 doses for other vaccine platforms. Statistically significant differences were observed in adverse event rates between mRNA vaccines and other vaccine platforms, except for one (Pfizer-BioNTech 79.05 vs CanSinoBIO 84.38, *p* > 0.05). The AE incidence rates were at least five times higher in recipients of the AZ from SII vaccine (474.59 AE per 100,000 doses) in comparison to other groups.

SAE rates were similar across all administered vaccines (range 0.72–1.61 AE per 100,000 doses), except for recipients of the AZ from SII vaccine (12.6 per 100,000 doses). All 32 SAE were hospitalized, of which 28 (87.5%) were discharged from the hospital and 4 cases died (12.5%). Documented deaths occurred among Pfizer-BioNTech [2], J&J [1], and Sinovac [1] recipients. Neurological AEs were the most common SAE reported followed by thrombotic events. A total of 7 (12.5%) cases of Guillain–Barré Syndrome (GBS) were reported across different brands (J&J, Sinovac, AZ from Oxford and Pfizer-BioNTech). The overall rate for GBS in our study was higher (0.19 per 100,000 doses) than the overall rate for three vaccine brands in a VAERS-based study (0.08 per 100,000 doses), but rates for GBS across J&J recipients were lower in our population (0.31 per 100,000 doses vs. 0.52 per 100,000 doses) [17]. Concerning thrombotic events, four cases of thrombosis related to vaccines were documented in our surveillance system: (1) Acute myocardial infarction (AZ/A1), (2) Cavernous sinus thrombosis (CVST), (PB/A1) (3) Ischemic stroke (PB/A1-C), and (4) Lower Extremity Thrombosis (SIN/A1), which represents a 0.1 rate per 100,000 doses in viral vector vaccines recipients and a 0.15 rate per 100,000 doses in mRNA vaccines recipients, both rates resulted lower than those reported in the literature [18,19]. Cerebral venous sinus thrombosis (CVST) with thrombocytopenia syndrome (TTS) has been described after the administration of several vaccines (Pfizer-BioNTech, AZ from Oxford, J&J, and Moderna), and a systematic review revealed that most of the reported cases occurred in females and 39% of cases died due to complications of CVST and VITT [20]. The causality evaluation of all severe AE revealed that most events (59.37%) were associated with the administered vaccine. These evaluations were carried out by a National Experts Committee in a case-by-case analysis and followed national guidelines.

Since the emergency vaccine authorization by the World Health Organization, concerns surrounding vaccine safety have been present around the global community despite multiple clinical trials reporting safety profiles as well as efficacy studies for different vaccines [8]. The use and application of these vaccines under emergency use by countries and regions requires close monitoring by various stakeholders to ensure their safety and effectiveness as reported by clinical trials [21].

Adverse events may go undetected during clinical trials due to the length of time of follow-up, making post-marketing surveillance or, in this case, post-emergency authorization crucial to providing evidence from the real world. Risk estimates derived from population-based surveillance systems can be interpreted as signals of new potentially causal associations or new aspects of known ones which may guide further verification actions in certain studies or groups after clinical trials [22]. Since the beginning of COVID-19 vaccination campaigns across the world, the European Medicine Agency (EMA) and the United States (US) Food and Drug Administration (FDA) have recognized approximately 30–40 different presentations of adverse reactions following vaccination for Pfizer-BioNTech’s and Moderna’s vaccines [23].

Concerning the mechanisms of adverse events following vaccination, there have been multiple that can account for observed AEs: (1) the most common can be attributed to the process of vaccination, ranging from a vagal reaction associated with anxiety to an inappropriate site of administration to infection due to unsafe management of the biologic; (2) concerning attenuated vaccines, reversion to a virulence state of organisms; (3) immune-mediated phenomena triggered mechanisms, such as IgE-mediated type I hypersensitivity reactions and immune-complexes (type III hypersensitivity reactions), can cause localized or systemic AEs; and (4) idiopathic and other autoimmune responses, such as thrombocytopenic purpura [24]. As for AEs of reported thrombotic events following vaccination against SARS-CoV-2 with certain viral vector vaccines (Oxford–AstraZeneca and Janssen), termed “vaccine-induced immune thrombotic thrombocytopenia”, these have been thought to be caused by the induction of antibodies against platelet factor 4 (PF4) by viral DNA and/or cellular proteins [25]. On the other hand, mRNA vaccines have been associated with anaphylactic shock above the average incidence in the population, particularly in those with a history of allergies. For the latter, polyethylene glycol (PEG) has been proposed as a potential causative agent [26]. In the case of myocarditis after SARS-CoV-2 mRNA vaccination, three main mechanisms by which vaccines might induce hyper-immunity have been proposed: (1) mRNA immune reactivity, (2) antibodies to SARS-CoV-2 spike glycoproteins cross-reacting with myocardial contractile proteins, and (3) hormonal differences [27]. Finally, mechanisms underlying GBS after SARS-CoV-2 vaccination have been the same ones described within the autoimmune known pathophysiology, and although large studies are lacking, recent works have documented the increased risk of GBS following the Janssen vaccination compared to mRNA vaccines [28].

Additionally, the present study adds to the evidence of vaccine safety in the Latino population, an underrepresented ethnic group in SARS-CoV-2 clinical trials [29], with only one trial including the Mexican population [30] before emergency authorization. As of May of 2022, only two research studies have documented vaccine safety across the country; a nationwide descriptive study focusing on neurologic adverse events among the BNT162b2 mRNA vaccine [31] and a comparative study on the extension of side effects among different vaccines recipients, which recruited participants via social media platforms [32].

Most of the AE that we documented in the present study are consistent with AE previously reported in clinical trials [30,33,34,35,36,37] and population-based reports [17,18,25,38,39,40,41,42,43] (Appendix A). In the case of clinical trials, we identified that the reporting of AE in published studies varied; while most reported AE and included a safety subgroup for detailed follow-up and unsolicited AE through active surveillance, differences were found between how AE were documented; only one study opted to monitor AE via passive surveillance [33], the rest included multiple groups; and the AZ from SII vaccine study had two cohorts, one for vaccine safety and a second one for evaluating immunogenicity/reactogenicity [30]. The J&J study documented unsolicited AE grade ≥ 3 during the following 28 days and established a safety net for solicited local and systemic symptoms [34]. The CanSinoBIO study reported AE from a safety net group, as well as serious and those defined as medically attended AE [30]. The AZ from Oxford vaccine trial reported all AE with severity distinctions [36], as well as those grades 3 or higher and non-serious AE. Finally, the clinical trial from Pfizer-BioNTech only reported adverse total adverse events by severity [37].

In the case of vaccines submitted for evaluation in the US, vaccine manufacturers were required to present safety data from Phase 1 and 2 studies for authorization, specifically focusing on serious adverse events, adverse events of special interest, as well as data from safety follow-up from Phase 3 studies, which are required to include a median follow-up duration of at least two months after the completion of the vaccination regime to assess a vaccine’s benefit–risk profile [44]. For post-marketing surveillance, the World Health Organization issued guidance on safety surveillance, specifically on criteria and definitions for AE and clinical developments. These guidelines include both adverse events following immunization (AEFI) and adverse events of special interest (AESI), and underlining that those countries unable to implement active surveillance for adverse events of special interest (AESI) should aim to report all AESI-like events as adverse events following immunization (AEFI) [21], which is the case of the Mexican AEFI surveillance system. It is important to comment that some AEFI described by the WHO are not included in Mexico’s epidemiological surveillance for AE, such as thrombocytopenia, acute myelitis, peripheral facial nerve palsy, vaccine-associated enhanced disease, multisystem inflammatory syndrome, and sensorineural hearing loss, among others [14]. A case can be made for events such as orofacial and oculofacial effects (facial, labial, and glossal edema) as well as temporary peripheral paralysis (Ex. Bell’s palsy), which have been associated with mRNA vaccines and are not included as diagnoses in the epidemiological reporting form [14,26]. Clinical diagnoses such as GBS, CVT, thrombosis, anaphylactic shock, and neurological diseases, have been previously reported during clinical trials [30,33,34,35,36,37] and after vaccine administration in population settings [17,18,25,38,39,40,41,42,43,45].

Further research is needed to understand the higher AE rates observed in AZ from SII recipients regarding manufacturing, storage, quality control, and distribution processes, which at this point could not be addressed. This was a finding that has not been described in the literature to date. Although rates were calculated based on applied doses as a common denominator, it is worth clarifying that the number of doses for this vaccine was the lowest (23,810 doses) compared to other applied vaccines (range 95,990–1,282,792 doses across five brands) and could prove to be a possible explanation for the observed differences. Despite these, severe and total AE rates were not higher than those reported in multiple clinical and population-based studies analyzing SAE rates from different vaccine brands (AZ from Oxford, Pfizer-BioNTech, J&J, and Moderna) [25,30,33,34,35,36,37,38,39,40,41,42,43,45] and those specifically describing AE from AZ-SII. When considering both AE from AZ from Oxford and AZ from SII, the total AE rate per 100,000 doses is 51.7, making it lower than most AE rates from other vaccine brands. Finally, it is of consideration to mention that multiple studies have reported data on AE after AZ-SII vaccination; this adds up to evidence concerning safety and reinforces the need to report AE by manufacturer vaccine, particularly in this case after technology transfer from AZ–Oxford to SII [46].

Regarding myocarditis after mRNA vaccines (both Pfizer-BionNTech and Moderna mRNA-1273), it is rare (0.3–5.0 cases per 100,000 vaccinated people) [27] although there is evidence stating a higher incidence after the second vaccination dose in adolescent males and young men. The U.S. Vaccine Adverse Event Reporting System (VAERS) lists 1136 cases (0.5 events per 100,000 doses) among 209,778,783 administered doses [47]. It is worth mentioning that myocarditis is not an AE included in Mexico’s surveillance study and thus does not allow for analysis, although overall, palpitations were reported in 0.008% of recipients of mRNA vaccines (PF) and 0.004% among recipients of viral vector and inactivated virus vaccines. No further information on the surveillance system could confirm the diagnosis [14].

Finally, the current work expands the literature on AE after COVID-19 vaccination, as it compares rates per 100,000 doses between six different vaccines in a population-based setting, which, as of our latest literature search, no studies had analyzed and reported data as in this work and no population-based studies were available for Latin American countries, although studies from Europe and North America have compared AE rates between fewer vaccine brands. Additionally, vaccines included in this study were not authorized by the European Medicines Agency, the Food and Drug Administration (USA), and Canada Health, such as CanSinoBIO, Sinovac, and AZ from SII [48,49].

Our study has several limitations. In contrast to strict follow-up during clinical trials, the passive nature of surveillance systems may tend to underreport the incidence rate of AE, either by clinicians or patients themselves. Our study findings may involve unknown or unverified AE, since these types of surveillance systems do not provide medically confirmed or valid diagnostic evidence for mild adverse events. Concerning associated variables to the occurrence of AE, analysis was limited to available variables within the surveillance system due to the lack of a standardized registry of relevant clinical variables such as comorbidities and medications. Other variables that are not known include current SARS-CoV-2 infections at the moment of vaccination and during the following days. Analysis by age groups and vaccine manufacturers, as well as the number of doses received by individuals was not possible due to the lack of access to a vaccine registry at the individual level. Finally, we are aware that the number of doses applied by manufacturer differed during the time of the study. This reflects the availability of vaccines in Mexico and the rapidly changing nature of the population-wide vaccination campaign in 2021 (Figure 1). Given these differences in the denominators, chance can contribute to explaining the observed results, nonetheless, our findings should be continuously monitored with rigorous pharmacovigilance.

## 5. Conclusions

The incidence rates of adverse events following COVID-19 vaccination, both mild and severe, differed among the six vaccine manufacturers in this study. It is noteworthy that recipients of the AZ vaccine from SII reported at least five times more AEs based on incidence rates per 100,000 doses in comparison to other vaccines, but did not surpass reported rates in other studies, although further research is required to address this finding. None of the applied vaccines had AE rates above those reported in clinical trials and population-based studies. Overall, symptoms and clinical diagnoses reported within the surveillance system were consistent with findings of clinical trials and population-based studies in the literature. Among more than 3 million administered doses, the most common severe AE was GBS and only one case of CSVT was documented. Myocarditis was not a clinical diagnosis included in Mexico’s study case for AE and thus represents an area of opportunity for improvements in the surveillance system. Age was associated with higher odds of experiencing a SAE. Additional research is needed to understand post-vaccine symptoms across different populations, including those from Mexico and Latin America.

## Figures and Tables

**Figure 1 vaccines-10-01196-f001:**
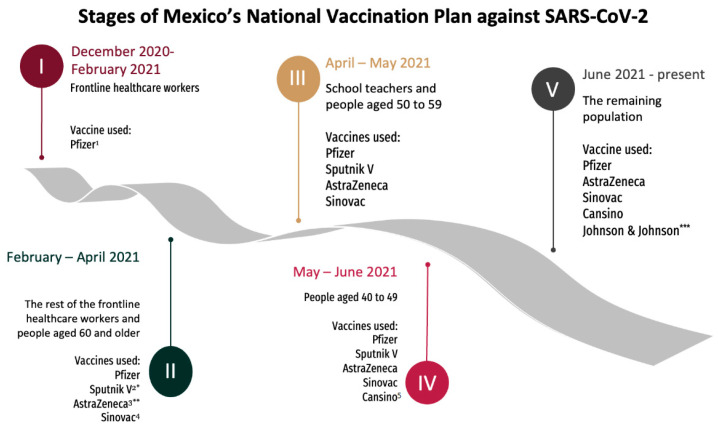
Vaccination strategy in Mexico. ^1^ BNT162b2 (Pfizer-BioNTech), ^2^ Gam-COVID-Vac (Gamaleya’s Sputnik V), ^3^ ChAdOx1 (AstraZeneca), ^4^ CoronaVac (Sinovac Life Sciences), and ^5^ Ad5nCoV (CanSino-BIO). Adapted from Política Nacional de Vacunación Contra el Virus SARS-CoV-2, Para la Prevención de la COVID-19 en México 2021 [11]. * The Sputnik V vaccine was not administered in the state of Baja-California. ** Includes a group of AstraZeneca vaccines manufactured at India’s Serum Institute (COVISHIELD) administered in Stage II in Baja-California. *** Johnson & Johnson vaccines were administered along the border in Stage V thanks to a donation from the Government of the United States of America Government as part of binational efforts to reopen the U.S.A.–Mexico border [16].

**Figure 2 vaccines-10-01196-f002:**
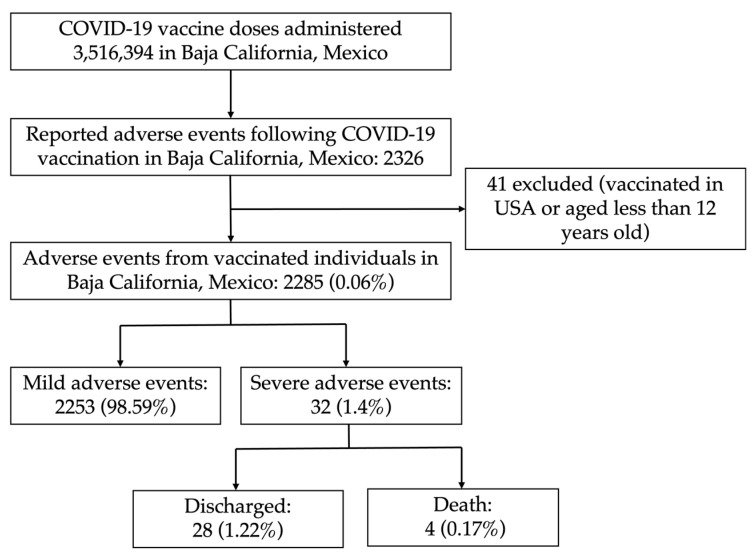
Adverse events following COVID-19 vaccination were reported through the state epidemiological surveillance system in Baja-California, Mexico.

**Table 1 vaccines-10-01196-t001:** Mild, severe, and total adverse events per 100,000 administered doses by vaccine manufacturer.

Vaccine Platform	Vaccine Manufacturer	Administered Doses	Total AE (*n*)	AE per 100,000 Doses	*p*-Value	Mild AE (*n*)	Mild AE per 100,000 Doses	Severe AE (*n*)	Severe AE per 100,000 Doses
mRNA	Pfizer-BioNTech	1,282,792	1014	79.05		1004	78.27	10	0.78
Viral vector	CanSinoBIO	959,990	81	84.38	0.61	80	83.34	1	1.04
	Johnson & Johnson	1,256,494	825	65.66	<0.0001	816	64.94	9	0.72
	AstraZeneca	608,547	214	35.71	<0.0001	209	34.34	5	0.82
	Serum Institute of India	23,810	113	474.59	<0.0001	110	461.99	3	12.6
Inactivated virus	Sinovac	248,761	38	15.28	<0.0001	34	13.67	4	1.61
	Total	3,516,394	2285	64.98	-	2253	64.07	32	0.91

**Table 2 vaccines-10-01196-t002:** Age, sex, and symptoms distribution by vaccine platform (viral vector and mRNA).

	Adverse Events (Mild and Severe)			
Variable	Total (*n* = 2285)	Viral Vector/Inactivated Virus (*n* = 1271)	mRNA (*n* = 1014)	*p*-Value
Age-mean ± SD	37.68 ±12.93	35.42 ± 12.98	40.5 ± 12.29	<0.0001
Gender				
Women—*n* (%)	1532 (67.05)	808 (63.57)	724 (71.40)	<0.0001
Pregnant—*n* (%)	29 (1.27)	20 (1.57)	9 (0.89)	0.11
History of allergic reactions—*n* (%)	326 (14.27)	153 (12.03)	173 (17.06)	<0.001
Food	45 (1.97)	14 (1.10)	31 (3.06)	<0.05
Drugs	181 (7.92)	81 (6.37)	100 (9.86)	<0.05
Pollen	26 (1.14)	11 (0.87)	15 (1.48)	0.23
Ignored	8 (0.35)	3 (0.24)	5 (0.49)	0.49
Other	25 (1.09)	15 (1.18)	10 (0.99)	0.81
No allergy	1959 (85.73)	1118 (87.96)	841 (82.94)	<0.001
Symptoms—*n* (%)				
Headache	1766 (77.29)	1042 (81.98)	724 (71.40)	<0.0001
Myalgias/arthralgias	1464 (64.07)	919 (72.31)	545 (53.75)	<0.0001
Fever/hyperthermia	1069 (46.78)	695 (54.68)	374 (36.88)	<0.0001
Chills/diaphoresis	959 (41.97)	631 (49.65)	328 (32.35)	<0.0001
Local injection site reactions	1212 (53.04)	610 (47.99)	602 (59.37)	<0.0001
Gastrointestinal symptoms *	1023 (44.77)	600 (47.21)	423 (41.72)	<0.01
Malaise/fatigue	748 (32.74)	399 (31.39)	349 (34.42)	0.21
Dizziness	680 (29.76)	360 (28.32)	320 (31.56)	0.11
Other **	358 (15.67)	165 (12.98)	193 (19.03)	<0.0001
Sore or scratchy throat	358 (15.67)	186 (14.63)	172 (16.96)	0.12
Cough	336 (14.70)	180 (14.16)	156 (15.38)	0.44
Rhinorrhea	342 (14.97)	169 (13.30)	173 (17.06)	<0.05
Palpitations (tachycardia)	211 (9.23)	101 (7.95)	110 (10.85)	<0.05
Dyspnea/difficult breathing	152 (6.65)	97 (7.63)	55 (5.42)	<0.05
Rash, generalized pruritus	180 (7.88)	91 (7.16)	89 (8.78)	0.2
Ocular manifestations	113 (4.98)	57 (4.48)	56 (5.52)	0.44
Neurologic symptoms (peripheral)	98 (4.29)	43 (3.38)	55 (5.42)	<0.05
Adenopathies/lymphadenopathies	62 (2.71)	19 (1.49)	43 (4.24)	<0.001

* Includes nine cases of intussusception. ** Includes those classified as other, pneumonia, osteoarticular lesions, hemorrhagic manifestations, movement limitations, syncope, and bronchial spasm.

**Table 3 vaccines-10-01196-t003:** A1: adverse event related to the vaccine, A2: adverse event related to a vaccine quality defect, A3: adverse event related to an operative or technical error, B: undetermined, C: causal association inconsistent with vaccination, D: causal association due to inherent characteristic of the vaccinated individual, E: unclassifiable.

Clinical Diagnosis	Time to AE (Hours)	Average Time (Hours)	Length of Stay (Days)	Outcome	Vaccine Platform	Vaccine Brand	Causality Evaluation
Acute Myocardial Infarction (*n* = 2)	1	20.33	-	Death	Viral Vector	Sinovac	D, E
	60		6	Discharged	Viral Vector	AstraZeneca	A1
Acute polyradiculopathy (*n* = 3)	0	0.72	14	Discharged	mRNA	Pfizer-BioNTech	A1
	2		4	Discharged	Viral Vector	SII PVT	A1, B
	0.16		-	Death	Viral Vector	Janssen	A1
Anaphylaxis (*n* = 1)	10	-	7	Discharged	mRNA	Pfizer-BioNTech	A1
Auricular fibrillation (*n* = 1) (arrhythmia)	7	-	3	Discharged	Viral Vector	Sinovac	D, E
Cardiopulmonary arrest (*n* = 1)	7.25	-	0	Death	mRNA	Pfizer-BioNTech	A1
Cavernous sinus thrombosis (*n* = 1)	0.08	-	16	Discharged	mRNA	Pfizer-BioNTech	A1, B
Encephalitis (*n* = 1)	9.16	-	5	Discharged	Viral Vector	Janssen	A1
Exfoliative dermatitis (*n* = 1)	0	-	6	Discharged	Viral Vector	Janssen	A1
Functional Diarrhea (*n* = 1)	6	-	7	Discharged	Viral Vector	CanSinoBIO	A1
Guillain–Barré Syndrome (*n* = 7)	0	4.5	6	Discharged	Viral Vector	Sinovac	D
	3		4	Discharged	mRNA	Pfizer-BioNTech	A1
	7.5		7	Discharged	Viral Vector	Janssen	B
	0		5	Discharged	Viral Vector	Janssen	A1
	21		-	Discharged	Viral Vector	Janssen	A1
	0		5	Discharged	Viral Vector	Janssen	A1
	0		10	Discharged	Viral Vector	AstraZeneca	C
Ischemic stroke (*n* = 1)	8.66	-	3	Discharged	mRNA	Pfizer-BioNTech	A1, C
Lower extremities thrombosis (*n* = 1)	0	-	6	Discharged	Viral Vector	Sinovac	A1
Lower right extremity acute neuropathy (*n* = 1)	1.5	-	3	Discharged	Viral Vector	Janssen	A1
Myelopathy (*n* = 1)	15	-	1	Discharged	Viral Vector	SII PVT	B
Neurologic deterioration and weakness (*n* = 1)	16	-	3	Discharged	Viral Vector	Janssen	A1, A3
Polyneuropathy (*n* = 1)	0	-	8	Discharged	mRNA	Pfizer-BioNTech	A1, B
Rhabdomyolysis (*n* = 1)	9	-	4	Discharged	mRNA	Pfizer-BioNTech	A1
Septic shock (*n* = 1)	0	-	2	Death	mRNA	Pfizer-BioNTech	A1
Stevens–Johnson Syndrome (*n* = 1)	48	-	-	Discharged	Viral Vector	AstraZeneca	A1
Transverse myelitis (*n* = 2)	15.91	8.21	15	Discharged	mRNA	Pfizer-BioNTech	A1
	0.5		7	Discharged	mRNA	Pfizer-BioNTech	B
Unspecified Radiculopathy (*n* = 1)	ND	ND	10	Discharged	Viral Vector	SII PVT	A1
Unspecified thrombocytopenia (*n* = 1)	23	-	11	Discharged	Viral Vector	AstraZeneca	A1

## Data Availability

The data presented in this study are available on request from the corresponding author.

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
