# Peer review of "Report of Adverse Effects Following Population-Wide COVID-19 Vaccination: A Comparative Study between Six Different Vaccines in Baja-California, Mexico"

_vaccines, 2022, doi:10.3390/vaccines10081196_

Round 1
Reviewer 1 Report
The title may be modified to read: "Report of adverse effects following population-wide COVID-19 vaccination: a comparative study between six different vaccines in Baja-California, Mexico"
A paragraph detailing the mechanisms underlying the adverse events reported would be most welcome.
Comparative data from other users of the same vaccines in different parts of the world can also be included as a table.
Whilel the methodology used of extrapolation of adverse events to a common denominator is widely accepted, the actual numbers of patients who were exposed to the vaccines from AstraZeneca and Serum Institute of India is indeed the low - to be frank, that from SII is the lowest.
Hence, an explanatory note as to the validity of the method used to extrapolate to a common denominator - i.e., 100 000 vaccinated - is purely a statistic and may not hold out could be included in the text.
Author Response
Point 1: The title may be modified to read: "Report of adverse effects following population-wide COVID-19 vaccination: a comparative study between six different vaccines in Baja-California, Mexico"
Answer: Thank you for the suggestion. The title has been modified according to suggestions from Reviewers 1 and 3.
Point 2: A paragraph detailing the mechanisms underlying the adverse events reported would be most welcome.
Answer: We appreciate the suggestion. Please find the proposed paragraph in the discussion section (Pages 9 and 10, starting in line 298).
Point 3: Comparative data from other users of the same vaccines in different parts of the world can also be included as a table.
Answer: The supplementary material now shows 2 tables comparing results from different clinical trials and population-based studies with the results from our surveillance system. These two tables are referenced in the main text in the Discussion section (page 10, line 335). Thank you for this valuable suggestion.
Point 4: While the methodology used for extrapolation of adverse events to a common denominator is widely accepted, the actual numbers of patients who were exposed to the vaccines from AstraZeneca and Serum Institute of India is indeed low - to be frank, that from SII is the lowest. Hence, an explanatory note as to the validity of the method used to extrapolate to a common denominator – i.e., 100 000 vaccinated – is purely a statistic and may not hold out could be included in the text.
Answer: We appreciate your insights on this matter. This point has been further discussed in the limitations of the study (Page 10, starting in line 373).
Reviewer 2 Report
This is an interesting paper, investigating adverse events (AEs) following the administration of six different vaccines in Baja-California, Mexico.
It is noteworthy the fact that the authors investigated the incidence and prevalence of AEs in an underrepresented populations in registration studies.
The main result was that AEs were at least five times higher in recipients of the AstraZeneca vaccine from the Serum Institute of India (AZ from SII).
Furthermore, AE rates were consistent with clinical trials and population-based reports in the literature, also in the investigated population.
However, in order to better understand and correctly interpret the results sowed in the manuscript, minor revisions are required.
Minor revisions:
· Table 1: What was the statistical test adopted? Does the p value refer to pairwise comparison? Is there a difference in the incidence of Mild and Severe AE among the different groups? The statistical analysis should be specified in the table caption.
Author Response
Table 1: What was the statistical test adopted? Does the p-value refer to pairwise comparison? Is there a difference in the incidence of Mild and Severe AE among the different groups? The statistical analysis should be specified in the table caption.
Answer: Thank you for the suggestion. We specified the statistical analysis in the table caption. Basically, the AE rate of each vaccine was compared against the Pfizer vaccine, which was the point of comparison for most of the explorative comparative analyses throughout the manuscript. Concerning differences between mild and severe AE; Mild AE yielded comparable p-values with those shown for total AE rates presented in the table. Due to the very small sample size of severe AE, we decided not to include calculated p-values.
Reviewer 3 Report
The article submitted by Mendez-Lizarraga et al to Vaccines is on a very relevant topic. However, I have very serious concerns about the data interpretation.
(i) Though the data is normalized to 100,000 doses of vaccinated groups, the number of people who received AstraZeneca from SII was very low as compared to the other vaccine groups. The group of people who received this vaccination needs to be followed up on what background they received AZ (SII). I mean the demographic characteristics and health conditions needs to be re-checked and make sure they were healthy normal, if you specifically mention it has higher rates of adverse effects-relative to lower doses received by the group.
(ii) Why the AstraZeneca vaccine was alone subcategorized based on manufacturer. Why the data was not combined or is there a difference between the two vaccines from Oxford and SII. If so, please mention in the article.
(iv) If we are specifically telling that the vaccine “X” created higher rates of adverse effects, then the data should clearly specify in each section that you studied. For eg: Result 3.1 clearly shows the vaccine platform, vaccine manufacturer and number of MAE and SAE with total number of administered doses. For the rest of the data Results 3.2 and 3.3 the data was not subcategorized based on vaccine manufacturer, which is a very important concern. There was no death reported for AZ (SII)
v) If you combine the AZ (Oxford and SII) together, the MAE per 100,000 doses is only about 50.44 and SAE about 1.27 which is lower than other groups.
I would suggest reconsidering the title change or organize data to make a strong view on the topic.
Author Response
(i) Though the data is normalized to 100,000 doses of vaccinated groups, the number of people who received AstraZeneca from SII was very low as compared to the other vaccine groups. The group of people who received this vaccination needs to be followed up on what background they received AZ (SII). I mean the demographic characteristics and health conditions need to be re-checked and make sure they were healthy normal, if you specifically mention it has higher rates of adverse effects-relative to lower doses received by the group.
Answer: We appreciate your valuable insights on this matter. Although the proposed is the best course of action to take, unfortunately there was no follow-up as part of the methodology presented in this study, and the investigators for this study do not have access to a nominal registry of all vaccinated individuals (denominators), only to the people who presented adverse effects (numerators). Based on the concerns brought by Reviewer 3, we did a post hoc analysis comparing the demographic characteristics of the people who developed AE from the SII group with patients presenting AE in the rest of the vaccines. These analyses showed differences in age means between AZ from SII vs the rest of the administered vaccines (p <0.0001), as seen in the Supplementary Table 3.
Also, we included a discussion on this matter in the limitations of the study in the discussion section, which mentions the lack of access to a nominal registry of all vaccinated individuals.
(ii) Why the AstraZeneca vaccine was alone subcategorized based on manufacturer. Why the data was not combined or is there a difference between the two vaccines from Oxford and SII. If so, please mention in the article.
Answer: For the same very reason clinical trials were designed and carried out specifically for the vaccine made by SII we decided that it was best to differentiate AE rates between manufacturers. Also, federal health authorities of epidemiological surveillance established these categories based on the manufacturer for surveillance purposes. You can find these in the discussion section, paragraph ten.
(iv) If we are specifically telling that the vaccine “X” created higher rates of adverse effects, then the data should clearly specify in each section that you studied. For eg: Result 3.1 clearly shows the vaccine platform, vaccine manufacturer and number of MAE and SAE with total number of administered doses. For the rest of the data Results 3.2 and 3.3 the data was not subcategorized based on vaccine manufacturer, which is a very important concern. There was no death reported for AZ (SII)
Answer: We agree with the statement, nevertheless, as outlined in the statistical analysis section, exploratory analyses were made to evaluate differences between vaccine platforms, based on the fact that mRNA vaccines were not previously employed on such a large scale (table 1). Secondly, differences in the number of administered doses vary greatly and would not be, statistically speaking, the best way to compare such a large sample (eg, mRNA recipients) vs very small sample (other manufacturers) which is why reported symptoms were categorized in two large groups, mRNA recipients (new technology) vs viral vector & inactivated vaccines (previously used technologies) in table 2. Lastly, we would like to clarify that severe adverse events are subcategorized based on their clinical presentation and are now described based on the vaccine brand within the text, (Discussion section, 3rd paragraph). The Discussion section also emphasizes the fact that no deaths for the AZ from SII were reported.
v) If you combine the AZ (Oxford and SII) together, the MAE per 100,000 doses is only about 50.44 and SAE about 1.27 which is lower than other groups.
Answer: Thank you for pointing this out. This has been included within the text (see paragraph ten in the discussion section).
vi) I would suggest reconsidering the title change or organize data to make a strong view on the topic.
Answer: Thank you for the suggestion. The title has been changed to better reflect the scope of the study as Reviewers 1 and 3 suggested.
Round 2
